# Broadband nonlinear modulation of incoherent light using a transparent optoelectronic neuron array

Dehui Zhang[1], Dong Xu[2], Yuhang Li[3], Yi Luo[3], Jingtian Hu[3], Jingxuan Zhou[2], Yucheng Zhang[2], Boxuan Zhou[2], Peiqi Wang[1], Xurong Li [3], Bijie Bai[3], Huaying Ren [1], Laiyuan Wang [1], Ao Zhang [2], Mona Jarrahi [3,4], Yu Huang [2,4], Aydogan Ozcan [3,4] ✉ & Xiangfeng Duan [1,4] ✉

Nonlinear optical processing of ambient natural light is highly desired for computational imaging and sensing. Strong optical nonlinear response under weak broadband incoherent light is essential for this purpose. By merging 2D transparent phototransistors (TPTs) with liquid crystal (LC) modulators, we create an optoelectronic neuron array that allows self-amplitude modulation of spatially incoherent light, achieving a large nonlinear contrast over a broad spectrum at orders-of-magnitude lower intensity than achievable in most optical nonlinear materials. We fabricated a 10,000-pixel array of optoelectronic neurons, and experimentally demonstrated an intelligent imaging system that instantly attenuates intense glares while retaining the weaker-intensity objects captured by a cellphone camera. This intelligent glare-reduction is important for various imaging applications, including autonomous driving, machine vision, and security cameras. The rapid nonlinear processing of incoherent broadband light might also find applications in optical computing, where nonlinear activation functions for ambient light conditions are highly sought.

General-purpose optical computing[1–5] and computational imaging[6,7] would significantly benefit from optical nonlinearity[8–17]. For various practical applications involving the processing of visual information and scenes, the nonlinear optical layer must operate at low optical intensities and high frame rates, covering spatially incoherent broadband illumination. Furthermore, energy loss through the nonlinear layers should be minimal to preserve the information carried by the optical fields. These demanding requirements on high speed, large nonlinear coefficient, low threshold intensity, low loss, and broadband response can hardly be achieved with existing nonlinear optical materials (Supplementary Table 1). For example, the ambient light intensity captured by digital cameras is typically below -0.1 W/cm² (refs. 18,19), which is many orders of magnitude weaker than the intensities employed in typical nonlinear optical processes, such as second harmonic generation (>10¹¹ W/cm²)[20–22], ultrafast nonlinear absorption (>10⁶ W/cm²)[23,24], and nonlinear Kerr effect (>10⁴ W/cm²)[25,26]. Alternative forms of nonlinearity, such as photochromic[27,28] and photorefractive[29–31] effects, may also be used, but these processes are typically slow, involving response time scales on the order of a few[27,29] to tens of seconds[31]. Other forms of photorefractive effect, for example, in multiple quantum wells, are fast but present very weak nonlinearity[32–34]. Some organic photorefractive devices show millisecond responses, but rely on coherent interference to build up spatial charge patterns and create strong nonlinear diffraction patterns, and are thus mostly adapted to holography[35,36]. Many of these photochromic or photorefractive devices also feature strong absorption (e.g., >90%),

[1]Department of Chemistry and Biochemistry, University of California, Los Angeles, CA, USA. [2]Department of Materials Science and Engineering, University of California, Los Angeles, CA, USA. [3]Department of Electrical and Computer Engineering, University of California, Los Angeles, CA, USA. [4]California NanoSystems Institute (CNSI), University of California, Los Angeles, CA, USA. ✉e-mail: ozcan@ucla.edu; xduan@chem.ucla.edu

leading to substantial optical losses[37]. Additionally, most of the existing nonlinear optical processes work for only a limited wavelength range.

Ultimately, the challenge lies in the fundamental tradeoff among power, speed, and transparency. Strong nonlinear responses require significant photo-induced physical/chemical changes in the material. This is achieved in typical photochromatic/photorefractive devices by either absorbing a sufficient number of photons in a short time interval, which either requires intense illumination far beyond the intensity of natural light or strong optical absorption that leads to low transparency and substantial losses[37,38]; or capturing a smaller portion of photons and gradually accumulating the photo-induced changes over a longer period, which is more transparent but with an intrinsically slow response[27,31]. Addressing these challenges requires a fundamentally distinct working mechanism that can use a small amount of optical power to strongly modulate the incoming photons.

Herein, we report a new strategy using an optoelectronic neuron array to achieve strong optical nonlinearity at low optical intensity, enabling broadband nonlinear modulation of incoherent light. This nonlinear optoelectronic neuron array is created by the heterogeneous integration of two-dimensional (2D) transparent phototransistors (TPTs)[39–41] with liquid crystal (LC) modulators. The designed optoelectronic neurons allow spatially and temporally incoherent light in the visible wavelengths to tune its own amplitude with only ~20% photon loss. For a proof-of-concept demonstration, we fabricated a 100 × 100 (10,000) optoelectronic neuron array, and demonstrated a strong nonlinear behavior under laser and white light illumination. The nonlinear filter-array was further integrated as part of a cellphone-based imaging system for intelligent glare reduction, selectively blocking intense glares with little attenuation of the weaker-intensity objects within the field of view. Our device modeling suggests a very low optical intensity threshold of $56\,\mu W/cm^2$ to generate a significant nonlinear response, and a low energy consumption of 69 fJ per photonic activation after proper device optimization. Our results demonstrate a new type of 2D optoelectronic device for the nonlinear processing of low-power broadband incoherent light, which is highly desirable for image processing and visual computing systems that do not rely on intense laser inputs. When coupled with free-space optical computing systems, composed of, for example, task-specific diffractive processors[4,42–44], the optical inputs to this nonlinear optoelectronic neuron array can take programmable convolutional or fully-connected filters, covering any desired set of spatially varying point spread functions between the visual scene and the plane of the optoelectronic device. Therefore, cascaded integration of optoelectronic neuron arrays with linear diffractive processors could be used to construct nonlinear optical networks, potentially finding widespread applications in computational imaging and sensing, and also opening the door for new nonlinear optical processor designs using ambient light.

## Results

### Device design, fabrication, and operation principles

Our proof-of-concept device consists of a 10,000-pixel array of parallel connected optoelectronic neurons (Fig. 1a–c). Each pixel consists of an independent 2D TPT as the photo-gate, and an LC layer simultaneously as the load resistor and the optical modulator, forming an optoelectronic neuron (Fig. 1d, e). The desirable neuron performance (nonlinear modulation) was achieved through a distinct working mechanism using local optoelectronic logic feedback to achieve intelligent modulation of the transmitted optical signal. The internal photoconductive gain allows a large change in the TPT resistance under low illumination ($0.1–10\,mW/cm^2$). The controllable and optimizable resistance match of the TPT and LC layer converts this resistance change to a large change in the voltage drop across the LC layer, which enables strong nonlinear behavior in the LC modulators. This heterogeneously integrated architecture allows independent

optimization of the TPTs and LC modules for desired performance metrics beyond the intrinsic tradeoff limits of a given material, such as in conventional photochromatic or photorefractive devices.

The TPTs were fabricated using solution-processed 2D van der Waals thin films (VDWTFs)[45,46], which are constituted by staggered 2D $MoS_2$ nanosheets (see Methods for details). With dangling-bond-free VDW interfaces among the staggered 2D nanosheets, the solution-processed VDWTF features excellent semiconductor properties[47,48] (see Supplementary Figs. 1 and 2 and Supplementary Table 2), which is essential for the required optical response and electrical operation frequency. The transparency of the VDWTFs can be tuned with controlled film thickness. The low-temperature solution process of ultra-thin $MoS_2$ VDWTF allows depositing the semiconducting film on arbitrary substrates at the wafer scale, offering an ideal material platform for making TPTs and heterogeneous integration with various optical modulators with few processing constraints.

We first prepared the $MoS_2$ VDWTF on a glass substrate, patterned the VDWTF into TPT channels, and then defined Cr/Au contacts. The TPT drain electrodes are connected to the supply voltage ($V_{dd}$), and the source electrodes are connected to a metal-ITO electrode stacked over the TPT pixel. An SU-8 layer isolates the ITO electrode from the TPT channel (except for the via) to remove the leakage current and parasitic capacitance. The top ITO electrode modulates a twisted nematic LC cell, which works both as a voltage divider and an optical modulator (Fig. 1e)[49,50]. Finally, a pair of orthogonal polarizers sandwich the TPT-LC stack (see Methods for details on the material preparation and device fabrication).

The 10,000-pixel optoelectronic neurons were connected in parallel, with the TPT terminals connected to $V_{dd}$ and the LC terminals connected to the common ground (GND). Despite this simplified circuit layout, each TPT has an independent optical gate (only controlled by the local illumination intensity on each pixel) that modulates the resistance of the individual TPT without significant electronic crosstalk from neighboring pixels. The change of the TPT resistance further modulates the voltage drop across the LC module in the specific pixel and thus the optical transmission of this pixel. The LC cell has a resistance of $R_{LC}$ ~ 10 GΩ per pixel (see Supplementary Note 1). We engineered the TPT channel to make its dark resistance higher than $R_{LC}$, and its light resistance below $R_{LC}$. Consequently, most of $V_{dd}$ falls on the TPT when it is dark, leaving the LC modulator transparent under rotating polarization (Fig. 1d, left panel). When the light is intense enough, most of $V_{dd}$ drops on the LC cell. The strong electric field rotates the LC molecules to suppress the optical transmission (Fig. 1d, right panel), resulting in a nonlinear transmission function. In this way, even though all TPTs and LC modules share the same $V_{dd}$ and GND, there is negligible crosstalk between individual pixels in terms of optical modulation. Each individual optoelectronic neuron in the 10,000-pixel array responds independently to local light intensity, but works in parallel concurrently without the need for electronic pixel scanning, which is essential for ensuring fast, low-power, and scalable operation.

### 2D transparent phototransistor array

We first prepared the VDWTF on a 10 cm × 10 cm glass substrate to demonstrate its uniformity and transparency over a large area (Fig. 2a). Figure 2b shows the TPT array under an optical microscope, which was fabricated using standard photolithography processes. The length and width of the TPT are $90\,\mu m \times 85\,\mu m$, with an array periodicity of $100\,\mu m$. The TPT layer exhibits a transmission of 88.5% at 550 nm and generally contributes to <20% decrease in transmission within the visible band (Fig. 2c). Figure 2d shows the photoresponse of a VDWTF TPT wired out for single-device tests, where we used a 633-nm laser as the light source. The beam was defocused with a spot diameter overlapping with the $100\,\mu m \times 100\,\mu m$ pixel area (see Methods). Our device design needs a large ON/OFF current ratio under illumination/darkness for the corresponding optoelectronic neuron to achieve

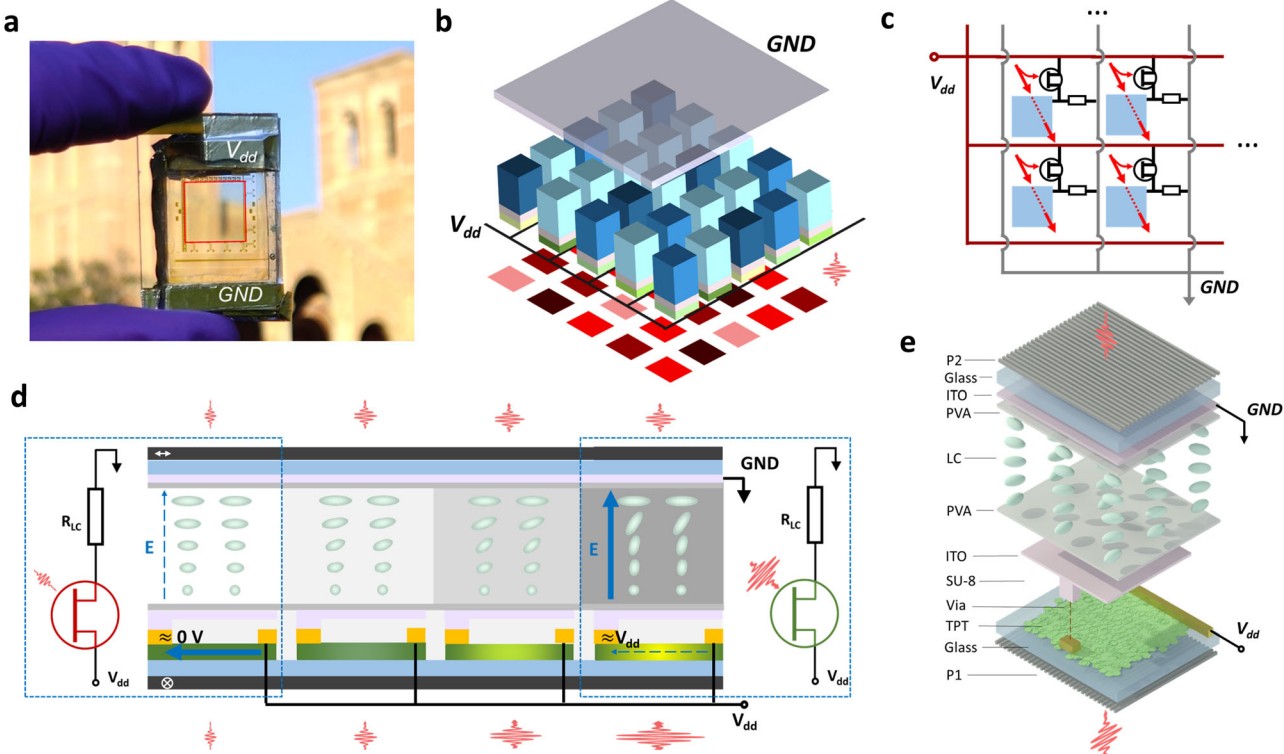

**Fig. 1 | Device configuration and working mechanism. a** A photo of the finished transparent optoelectronic neuron array (marked by the red box, polarizers not included). The 100 × 100-pixel array covers 1 cm × 1 cm area. **b** Diagram of an array with all neurons connected to the common $V_{dd}$ and GND electrodes. A globally wired metal electrode (black lines in Fig. 1b and gold wires in microscope images in Fig. 2b) is fabricated with lithography and applies $V_{dd}$ to all pixels in parallel, as also illustrated in Fig. 1c. Each neuron responds independently to the local illumination intensity (rendered as pink to dark brown input patterns) to provide the corresponding nonlinear transmissions. **c** The equivalent circuit of the neuron pixels sharing common $V_{dd}$ and GND. The red arrows represent the local incident light. **d** Self-amplitude modulation of light in the TPT-LC optoelectronic neuron array. Under low light illumination (left pixel), the TPT is highly resistive, most voltage drop occurs on the TPT. Thus, the voltage drop and electrical field across the LC are small (blue thin dashed line represents a weak field). The LC is unperturbed and remains transmissive. The equivalent circuit is shown on the left side, where the TPT is highly resistive and highlighted in red. At high input optical power, the TPT becomes conductive (marked in green in the equivalent circuit on the right side), so most voltage drops across the LC layer (thick solid line represents a strong field), shutting off the optical transmission. **e** Schematic illustration of a single-pixel structure disassembled by layers. The light is incident from the bottom, passing through the first polarizer (P1) and then through a glass substrate and a TPT made with $MoS_2$ VDWTF. An SU-8 insulating layer isolates an ITO electrode from the TPT channel. The ITO electrode is only locally connected to a single TPT and isolated from nearby pixels. The liquid molecules are anisotropic, rendered as ellipsoids. The molecules gradually rotate their 3D orientations to form a spiral pattern, rotating the incident light's polarization. A polyvinyl alcohol (PVA) alignment layer controls the LC molecule orientation at the PVA-LC interface. Another orthogonal alignment layer is on top of the LC layer, so the LC molecules gradually twist in the cell. Electrical current flows through the TPT to the ITO layer in the middle, then to the top ITO layer, which is grounded. The LC cell also works as an optically inactive resistor.

large nonlinearity. The channel current increases from 0.12 nA in the dark state to 1.96 nA at an input illumination intensity of 65.8 mW/cm², corresponding to an ON/OFF ratio of 16. This large resistance change under a moderate illumination intensity is essential for creating sufficient voltage modulation across the LC layer to produce a strong nonlinear transmission contrast (Supplementary Note 1).

Next, we used a chopper to modulate the light and study the photoresponse speed of our device at 488 nm and 633 nm wavelengths. These experiments show that the 3 dB cutoff frequencies are above 3 kHz for both wavelengths (Fig. 2e). Moreover, the responsivity at the red light is only slightly lower than that at the blue light because the bandgap of the few-layer $MoS_2$ is below 1.9 eV (>653 nm)[51,52]. We further characterized the photoresponse of 116 single pixels of individually wired TPTs (from four chips with identical process flow) under the illumination of a thermal lamp (at an intensity of ~21 mW/cm², see Methods), revealing a dark/light resistance ratio of 9.0 ± 1.3 (mean and standard deviation) (Fig. 2f), suggesting a relatively uniform photoresponse under broadband spatially incoherent light.

### Nonlinear self-amplitude modulation
We next evaluated the nonlinear amplitude modulation of the optoelectronic neurons by quantitatively analyzing the transmission of a

473-nm laser under varying incident intensity. The transmission ratios are normalized with respect to the high-transmission state (i.e., at $V_{dd} = 0$ V for Fig. 3a, see Methods). Our results showed that the transmission drops substantially with increasing illumination power, showing a modulation onset intensity of 0.1–10 mW/cm² (Fig. 3a). The exact onset power and modulation ratio can be readily tuned by varying the supply voltage ($V_{dd}$). An opposite nonlinear mode can also be achieved with the same device by rotating the two polarizers to be parallel to each other: in this case, the transmission ratio is low for dim light, but increases to near unity at high illumination power (see Fig. 3b). We further tested the nonlinearity under 650 nm laser (Fig. 3c) and white LED (Fig. 3d) illumination and observed similar nonlinear responses. The nonlinear onset for white LED illumination is ~1000 lux, equivalent to ~0.5 mW/cm², which is more than two orders of magnitude smaller than typical sunlight intensity[53] (Fig. 3d inset), demonstrating that the device is fully functional under low-intensity broadband spatially incoherent light. No transmission modulation was observed on the LC devices without the TPTs under the same test conditions, excluding a possible thermal contribution to the observed nonlinearity.

To evaluate the overall yield and uniformity of the optoelectronic neuron array, we examined the transmission at $V_{dd} = 6$ V and 14 V

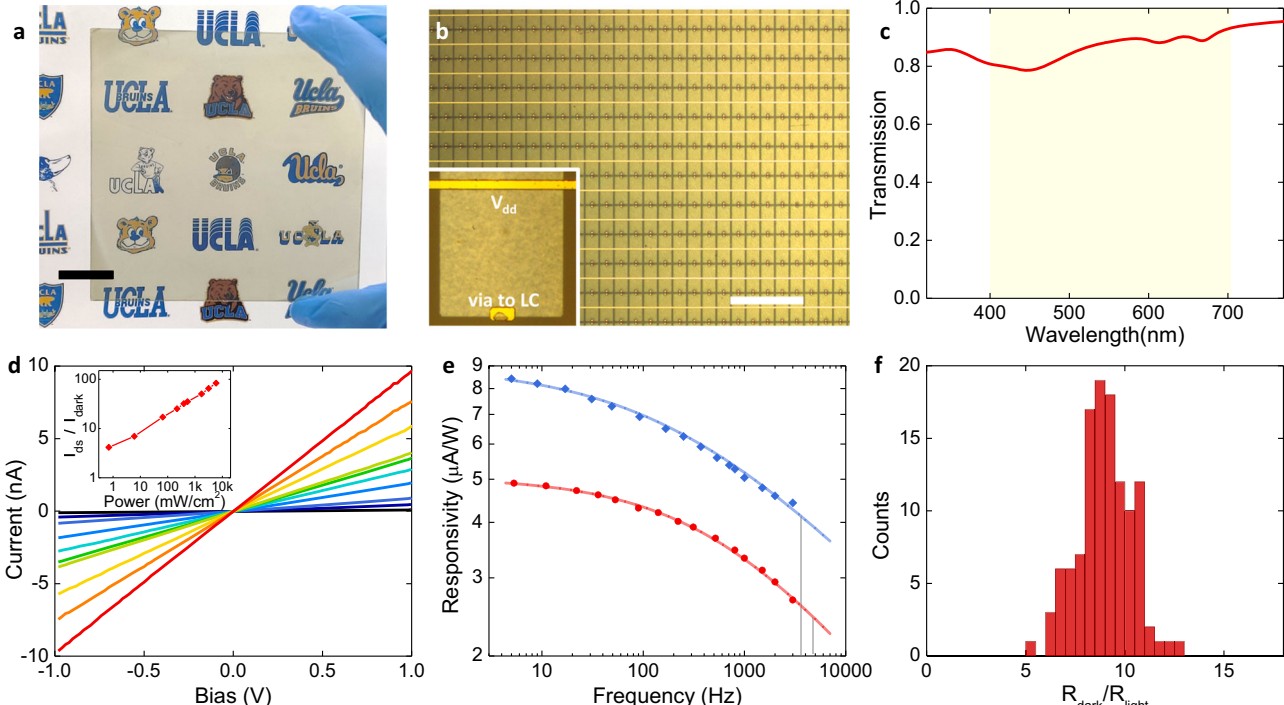

**Fig. 2 | Performance quantification of the TPTs. a** A photo of a 10 cm × 10 cm glass substrate coated with the MoS$_2$ VDWTF, demonstrating large-area scalability and transparency. Scale bar: 2 cm. **b** A microscopic image of the TPT array. The gold wires are connected directly to $V_{dd}$. The vias (the small gold spots) connect to the ITO electrode covering the entire pixel area. Scale bar: 500 μm. The contrast is large despite the high transparency of the device since the microscope works in reflection mode. **c** The transmission spectrum of the VDWTF in the visible range (highlighted in yellow shade). **d** The photocurrent of a separately wired out TPT at sweeping source-drain bias under 633-nm laser. Black to red (lower to higher): under the illumination intensity of 0, 0.7, 6.0, 65.8, 220, 393, 531, 1740, 3140, and 5920 mW/cm$^2$, respectively. Inset: the ratio of the channel current over dark current at different optical intensities. **e** The responsivity measured by a lock-in amplifier at different laser chopping frequencies. Blue: data from a 488-nm laser at 0.46 mW (4.6 W/cm$^2$) on a single pixel; red: data from a 633-nm laser at 0.59 mW (5.9 W/cm$^2$). Gray lines indicate the extrapolated 3 dB cutoff frequencies at 3.5 kHz and 4.7 kHz, respectively, which are above the chopping frequency available in our laboratory. **f** The ratio of the dark resistances over the light resistances of 116 TPTs, with a dark/light resistance ratio of 9.0 ± 1.3 (mean and standard deviation). The channel bias was 1 V for **e** and **f**.

under uniform, white LED illumination (Supplementary Fig. 4). The uniform ON/OFF transmittance suggests an essentially unity yield for 10,000 devices fully covered in the field of view. We further characterized the uniformity of the nonlinear response of 99 devices under different illumination levels with a histogram (Fig. 3d), demonstrating distinct transmission under different illumination fluxes for all devices.

### Integration of the optoelectronic neuron array with a cellphone camera for glare reduction

Next, we integrated our optoelectronic neuron array with a cellphone camera for intelligent glare reduction. Bright glares, such as sun glare and high beams during driving, stray laser beams in labs, or welding glares in factories, can trigger corneal reflexes or damage the human eye, as well as saturate image detectors in cameras, obscuring valuable information. Thus, bright glares are undesirable in various human/robot working conditions. Uniform attenuators or polarizers are used in sunglasses to solve this issue partially. However, such attenuators are linear optical elements and uniformly attenuate the glare and the working environment by the same level, and thus cannot selectively maintain useful information. Ideally, a glare-reduction imaging solution should selectively dim the bright rays while efficiently capturing the useful information from the surrounding, less-bright objects, which is highly desirable for many applications.

Our nonlinear optoelectronic neuron array offers an ideal solution to this challenge as it can filter out glares at a desired intensity threshold (that can be fine-tuned electronically using $V_{dd}$), operating at a broad visible spectrum faster than the typical video frame rate.

Therefore, it can intelligently suppress fast-changing glares over a large field of view with an electrically tunable intensity threshold. To showcase this capability, we constructed an imaging system using a cellphone camera integrated with our optoelectronic neuron array (see Fig. 4a and Supplementary Fig. 5). With this setup, we imaged grating-like object patterns under white light illumination with or without an intentionally generated intense glare (473 nm laser) to demonstrate intelligent glare reduction.

The patterns were clearly resolved when there were no glares (Fig. 4b). After turning on the glare while keeping the exposure and focus condition of the cellphone camera the same (Fig. 4c), the image was saturated by the intense scattered glare in the center, leaving little detectable features in the surrounding areas. Next, we applied $V_{dd}$ = 6 V to switch ON the optoelectronic neurons. This voltage was chosen to place the LC pixels right below the nonlinear transmission threshold for ambient light, and well above the threshold when intense glares turn on the TPTs. The intense glare beam pushed the illuminated optoelectronic neurons to the low-transmission state. As a result, the captured cellphone image shows a much dimmer glare, while faithfully resolving the grating patterns and recovering the spatial information (Fig. 4d) that was lost at $V_{dd}$ = 0 V (Fig. 4c). Direct transmission power measurement with a power meter shows the glare power drops by 84% when compared to its value at 0 V, while the non-glare region showed negligible transmission reduction.

Since most current camera systems often adopt auto-exposure software to adjust the exposure under different lighting conditions, we further compared the impact of our nonlinear filtering under auto-exposure settings. Turning on the auto-exposure of the cellphone

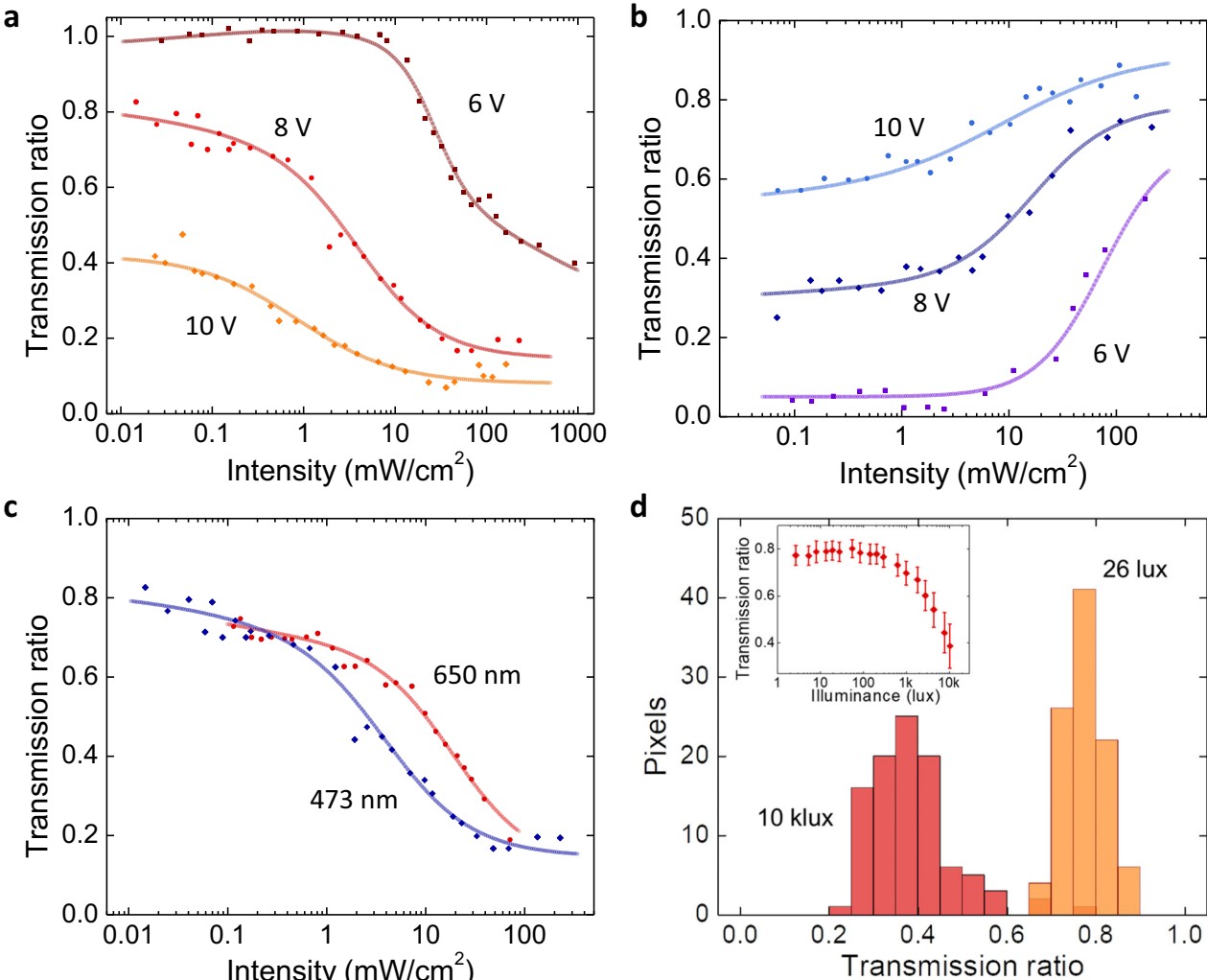

**Fig. 3 | Nonlinear transmission of the optoelectronic neuron array. a** The dependence of the transmission ratio on the incident power density of a 473 nm laser illumination under different voltage supplies. The two polarizers are orthogonal (see Fig. 1e). **b** Opposite nonlinear behavior at 473 nm when the two polarizers are parallel. **c** Nonlinear behavior with 650-nm laser illumination under orthogonal polarizer orientations at $V_{dd}$ = 8 V (red line). The result with 473 nm laser in panel **a** (blue line) is also plotted for comparison. **d** Histogram of the transmission ratio value distribution of 99 individual TPT-LC pixels at weak (orange, 0.77 ± 0.04, mean and standard deviation) and intense (red, 0.39 ± 0.09) white LED illumination (at $V_{dd}$ = 8 V with orthogonal polarizers). The transmission ratios are normalized by the transmission state at $V_{dd}$ = 0 V (see Methods). Inset: the mean value and standard deviation of transmission at different illuminance values. The incident laser in **a–c** was polarized along the polarizer (P1) direction. All data were measured on the same chip, with slight threshold deviations due to pixel-to-pixel variations.

camera without nonlinear filtering ($V_{dd}$ = 0 V) leads to a significant loss of the surrounding object patterns (Fig. 4e), which is due to the software-adjusted reduction of the image exposure resulting from the intense glare beams. Turning on the nonlinear filtering ($V_{dd}$ = 6 V) under the auto-exposure mode restores the patterns of the input scene as desired (Fig. 4f). These studies clearly highlight that our nonlinear filter can effectively reduce the negative impact of intense glares in a way that is not readily possible with software-corrected auto-exposure processes.

The glare degrades the image quality of its surrounding area in two ways: for regions close to the glare center (Box 2, 3 in Fig. 4b), the intense glare saturates the image; for regions slightly further away (Box 1, 4, 5), the irregular glare overlaps with the object patterns and significantly contributes to noise. Hence, we quantified the SNR and contrast of the acquired cellphone images to better highlight these observations (see the white boxes in Fig. 4b). We computed the average and standard deviation values along the grating lines (x-axis for Box 1 in Fig. 4b) to get the cross-sectional profiles perpendicular to the grating lines (Fig. 4b–f right five panels, see Methods for details). The noise increases by over ten times in the dark regions (see the error bars

for Box 1 in Fig. 4b, c) after turning on the glare. As expected, the contrast between the black and white stripes of the input object is substantially suppressed under the glare, leaving few resolvable features. Evaluated at Box 1 in Fig. 4b, the SNR (see Methods) is 14.2 for the no-glare case, which drops to 1.08 with the bright glare when the nonlinear filtering is off, and is restored to ~5.50 when the nonlinear filtering is turned on. Under the auto-exposure setting, the image SNR collected by the cellphone camera decreases to 0.86 due to the glare (without the nonlinear filtering), which increases to ~3.51 when the nonlinear filtering is on, thanks to the intelligent suppression of the glare by the optoelectronic neuron array. The SNR and contrast over all five selected boxes show considerably enhanced SNR and pattern contrast over different regions when the optoelectronic neurons were switched on (Supplementary Fig. 7).

We also evaluated the response speed of our optoelectronic neurons in reducing dynamic glares by taking videos with the laser-induced glare frequently switched on and off, and observed a similar glare reduction performance under dynamic glare conditions (see Supplementary Video 1). If the optoelectronic neurons were to have a

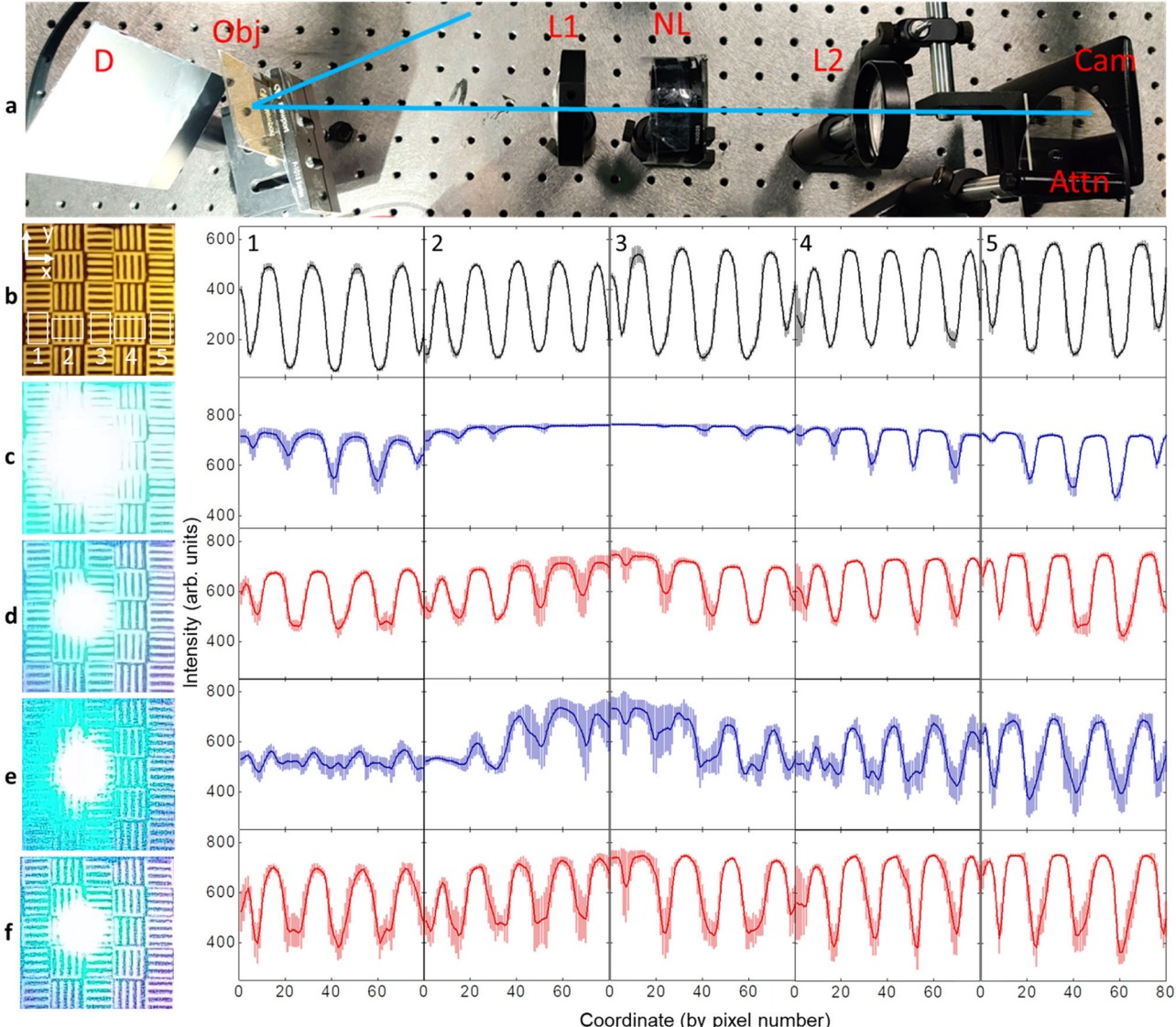

**Fig. 4 | Glare reduction with the nonlinear optoelectronic neuron array. a** The optical setup for the glare reduction test. White light from an LED array is diffusely reflected by an unpolished aluminum foil diffuser (D) and transmits through a patterned gold mask as the object (Obj). The pattern is imaged by the lens L1 on the image plane that overlaps with the optoelectronic neuron layer (NL). The light then passes through L2, which works as a magnifying system to adjust the field of view and the depth of field at the cellphone camera (Cam). An attenuator (Attn) adjusts the captured light intensity. The laser (blue ray) is reflected by the gold mask to create a bright glare. **b** Left: the raw image taken by the cellphone without glare at $V_{dd} = 0$ V. The average intensity (solid lines) and the noise levels (error bars) in the white boxes are plotted in the five figures on the right. **c** Results with laser-induced glare with a total glare power of 100 μW, measured by placing a power meter in front of L1. The camera settings are identical to **b**, $V_{dd} = 0$ V. **d** results with glare, $V_{dd} = 6$ V. **e** Results with glare. The auto-exposure of the cellphone camera is turned on. $V_{dd} = 0$ V. **f** Results with glare. The auto-exposure is turned on. $V_{dd} = 6$ V.

response speed slower than the video frame rate (60 Hz), we would expect a delayed transmission change, leading to a dark spot at the glare center right after the glare is shut off or an overexposure when the glare is just turned on. Importantly, no such phenomenon was observed even in the first frame after the laser was turned off (Fig. 5a) or on (Fig. 5d). We further estimated the glare reduction response speed by calculating the average intensity of the displayed area and the selected red box region (in Fig. 5a). The intensity temporal evolution showed an instant switch within the first frame after the glare was off (Fig. 5a, right panel), suggesting a faster response than the video frame rate of 60 Hz (~17 ms). Comparing the dynamic video frame evolution when the optoelectronic neurons are on (Fig. 5a) with that when they are off (Fig. 5b), it is apparent that the video obtained with the optoelectronic neurons on shows a clear benefit when there is glare, and preserves the same image quality starting with the first frame after the glare is turned off, further confirming the instant response of the

optoelectronic neuron array. In contrast, with the software-controlled auto-exposure, the cellphone camera showed a series of darker frames initially when the glare is suddenly turned off, and did not reach a normal exposure condition until beyond 1 s (Fig. 5c), due to the delayed response from the software-adjusted exposure reduction when the glare was on.

Similarly, when the laser glare was turned on, our optoelectronic neuron array showed an instant response to achieve a significantly improved video quality (Fig. 5d) over that obtained with neurons off (Fig. 5e); in contrast, the software-adjusted image exposure takes more than 1 second to stabilize (Fig. 5f). The software-controlled auto-exposure correction is much slower because it needs multiple frames to adjust the image exposure, which can take seconds depending on the algorithm, computing hardware and illumination conditions[54]. These analyses and Fig. 5 demonstrate that our optoelectronic neuron array can rapidly respond to dynamically varying illumination conditions.

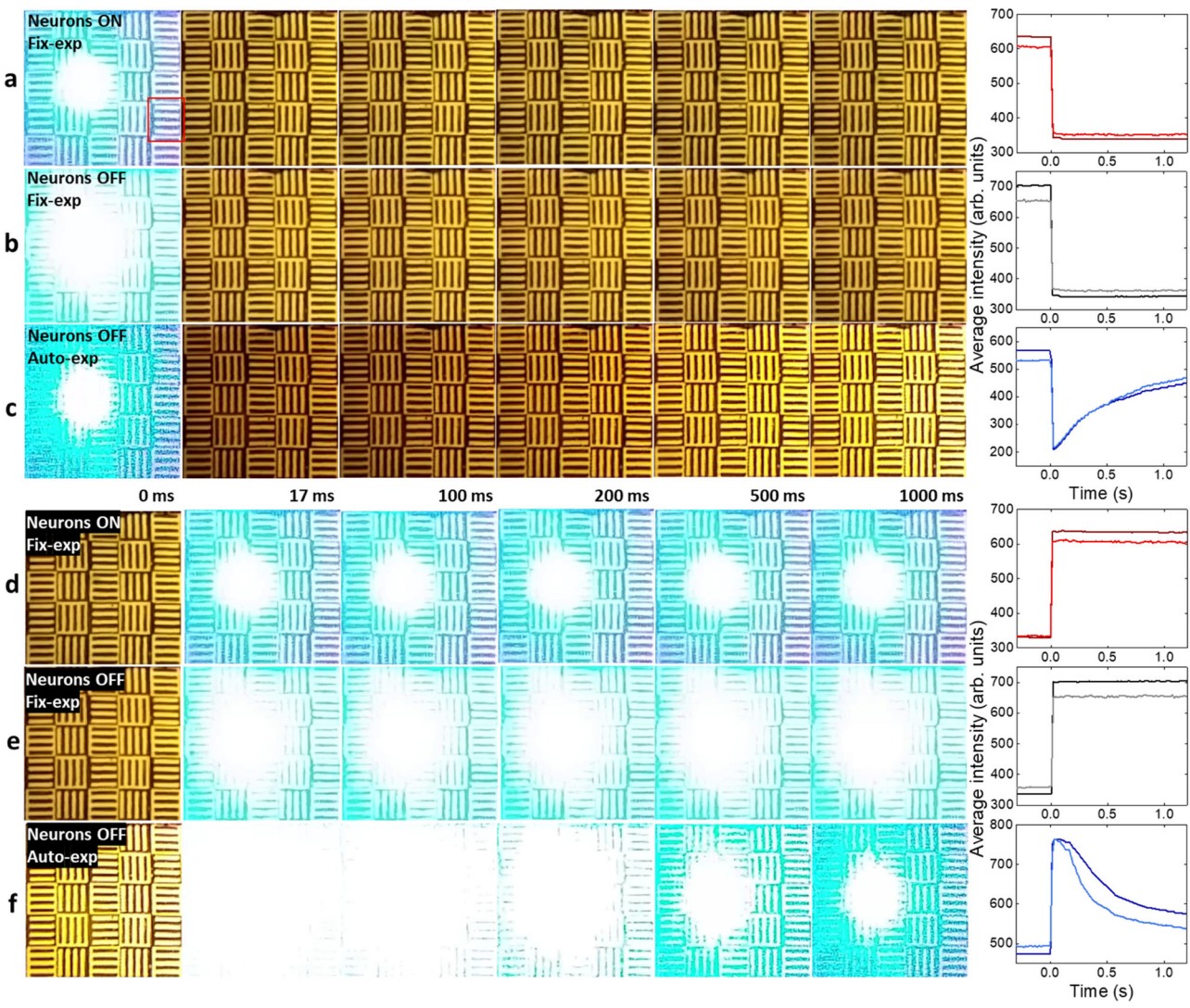

**Fig. 5 | Temporal response of the optoelectronic neuron array and software auto-exposure correction.** **a** Optoelectronic neuron at $V_{dd}$ = 6 V with a fixed exposure. The laser glare is suddenly turned off between the first frame (0 ms) and the second frame (~17 ms). **b** Frames with $V_{dd}$ = 0 V and fixed exposure as the glare is turned off between the first two frames. **c** Frames with $V_{dd}$ = 0 V and auto-exposure. **d** Frames with $V_{dd}$ = 6 V and fixed exposure as the laser glare is turned on between the first two frames. **e** Frames with $V_{dd}$ = 0 V and fixed exposure. **f** Frames with $V_{dd}$ = 0 V and auto-exposure. Curves on the right: the intensity averaged over all pixels in the displayed area (darker lines) and averaged in the red box area shown in a (lighter lines) of each frame, plotted with respect to time. The glare power and optical setup are identical to Fig. 4.

## Discussion

We have reported a unique strategy to create an optoelectronic neuron array for broadband nonlinear modulation of spatially incoherent light. Beyond the nonlinearity and response speed, additional figures of merit should be considered for practical applications: the threshold optical intensity for the nonlinear response, and the total power consumption (optical absorption plus the electrical power consumption). Our experimental device has a nonlinear intensity threshold of ~1 mW/cm² for broadband visible spectrum (Fig. 3d). It absorbs ~20% of the incident photons in the TPT layer, corresponding to a photon power loss of 20 nW/neuron. The electrical power consumption is 1.8 nW/neuron with the typical nonlinear operation voltage at 6 V and a current density of 3 µA/cm². Thus, the total power consumed by our experimental device is ~22 nW/neuron. Dynamic changes in the input light field could also lead to frequent charging/discharging of the LCs, but do not significantly contribute to the total power consumption (see Supplementary Note 1).

We believe the performance of our proof-of-concept demonstrations is largely limited by suboptimal material processing and device fabrication protocols (particularly related to LCs), and could be further improved by developing and adopting mature industrial processes[55-59]. Thanks to the heterogeneous integration of 2D TPT and LC modulators, the TPT and LC layers can be optimized separately without tradeoffs. For example, the LC modulators in our studies show a gradual modulation slope, requiring a $\Delta V$ ~ 2 V to switch from the high-transmission state to the low-transmission state (Supplementary Fig. 3a). In contrast, the state-of-the-art LC modulators can achieve much steeper slopes, with a much smaller switching voltage $\Delta V$ < 0.3 V[60,61]. This means a smaller resistance change in TPTs can trigger the LC modulators. Another possible optimization is the lowest transmission value of LC, which relies on high-quality LC alignment layers and polarizers. Commercial LC-based displays can achieve a contrast ratio beyond 5000, compared to ~10 in our proof-of-concept LC modulators. This would correspond to a 500-fold stronger glare reduction when the industry-quality LC modulators are integrated with our optoelectronic neuron arrays. To shed more light on this opportunity, we modeled the device performance with optimized LC parameters and a better match between the TPT ON/OFF resistances and

the LC resistance (see the discussion in Supplementary Note 1). This modeling indicates that the ultimate optical intensity threshold for ON/OFF switching can go below 56 $\mu W/cm^2$ for 633-nm illumination wavelength. The total power consumption per neuron scales linearly with the pixel area, and thus can be reduced substantially to 6.9 pW/neuron (27.7 $\mu W/cm^2$) by scaling the pixel size down to 5 $\mu m$, as proved feasible in commercial LCD technologies. The LC we used (4-cyano-4′-pentylbiphenyl, common name: 5CB) supports a modulation speed above 100 Hz[62]. Taking this modest estimation of 100 Hz as the device speed, the total power consumption can be calculated as 69 fJ per nonlinear activation. Even lower power consumption is possible with further material and device-level optimizations of LCs to increase the speed[63,64] and scale down the device footprint[65].

We should also note that there is some non-uniformity with our current optoelectronic neuron array. A calibration may be conducted for more precise use of these nonlinear neurons for optical computing and computational imaging. In this case, the nonlinear optical threshold variations among the neurons of a given optoelectronic array could be measured and calibrated so that computer simulations or in-situ training[66–68] can be optimized to best exploit the optical nonlinearity of each neuron regardless of the device-to-device variations.

In summary, we heterogeneously integrated 2D TPTs with LC modulators to form a 10,000-pixel nonlinear optoelectronic neuron array. Our design enables nonlinear self-amplitude modulation of spatially incoherent light, featuring a low optical intensity threshold, strong nonlinear contrast, broad spectral response, faster speed than video frame rate, and low photon loss. We further integrated this optoelectronic neuron array into a cellphone-based imaging system and demonstrated intelligent glare reduction by selectively attenuating bright glares while preserving the surrounding object information that is otherwise unrecognizable by the camera. This device might find various applications in autonomous driving, photography, and security cameras. Our nonlinear optoelectronic neuron array could find broad uses in computational imaging and sensing fields, and opens the door for new nonlinear processor designs for optical computing using ambient light that is spatially and temporally incoherent.

## Methods
### MoS₂ ink preparation
The raw molybdenite flake was electro-chemically intercalated by tetraheptylammonium bromide (THAB). 200 mg THAB was dissolved into 40 ml acetonitrile. The molybdenite was fixed on a copper electrode, which connected to the cathode of the power source. The anode was connected to a carbon electrode. The current was ~5 mA at a voltage of ~8.5 V. After 1-hour intercalation, the flake was thickened, and the intercalated layers looked dark green. We cut the intercalated molybdenite and ground the pieces into small flakes with a glass rod. We dissolved 900 mg polyvinylpyrrolidone into 50 mL dimethylformamide (DMF) and mixed the intercalated molybdenite into it. Next, we ultrasonicated the mixture for 1 hr. The whole solution looked dark green with small shining flakes.

The mixture was centrifuged under 12,100 rpm (9985 $g$, with $g$ as the gravitational acceleration on earth) for 15 min to remove DMF. After that, we poured the clear upper solution out and added 40 mL of isopropyl alcohol (IPA). The mixture was ultrasonicated for a second time until no sediment was visible. We repeated this process two more times to fully remove DMF. To concentrate the solution, we centrifuged it under 12,100 rpm for 20 min, then added 5–10 mL IPA (depending on the amount of intercalated flakes). We ultrasonicated the ink until there was no sediment.

We calibrated the ink concentration by dissolving 20 $\mu L$ ink into 1 mL IPA and measuring the UV-vis absorption (Beckman Coulter DU800 spectrophotometer). The peak should be ~380 nm with an absorption of ~1.7 for repeatable ink concentration. Finally, the calibrated ink was centrifuged at 3000 rpm (2500 $g$) for 3 min to remove

the remaining large flakes. We repeated this process five times, after which the ink was ready for use.

### Device fabrication
We used microscope slides as the transparent glass substrate. The substrate was first treated with oxygen plasma and becomes hydrophilic. We spin-coated the MoS₂ ink on the substrate at 2000 rpm for 30 s on a spinner. The spin-coating was repeated six times to achieve a film thickness of ~10 nm. The MoS₂ VDWTF was then annealed in a furnace at 400 °C for an hour in an Ar atmosphere.

Next, we patterned the MoS₂ channel and defined the TPT pixels as discussed in the paper. We evaporated HMDS on the sample for improved surface adhesion, then spin-coated SU-8 2005 photoresist on the device at 2000 rpm for 30 sec. The photoresist was pre-baked at 65 °C for 3 min, 95 °C for 3 min, and then 65 °C for 2 min, with the slow temperature change to reduce internal strain and peeling-off. Next, we exposed the SU-8 with lithography (Karl Suss MA6) for 20 s to crosslink the polymer. A post-bake process identical to the pre-bake was conducted before developing with SU-8 developer for 30 seconds. The developed sample was rinsed with IPA. The lithography process opened windows on the external contact pads and the via region for connecting the TPT to the ITO pad on top. After the development, we further annealed the sample at 150 °C in Ar for 30 min, with slow increase and decrease of temperature to avoid SU-8 peeling. This further annealing improves the SU-8 crosslinking and makes it more robust to subsequent thermal and liquid processes. The SU-8 thickness was 8 $\mu m$ measured by a surface profiler. It serves as a passivation for the TPT layer, as well as a thick insulating layer that reduces the parasitic capacitance between TPT and the top ITO pad.

Next, we sputtered a 50-nm ITO layer on the top of the sample. Sputtered ITO has good step coverage, sufficient to connect the bottom metal pad to the upper electrode area over the SU-8 step. The ITO was then patterned with lithography and wet etched with 1% hydrochloric acid to make the ITO pads electrically insulated from each other.

The PVA layer was prepared by first dissolving 5% weight of PVA (Sigma Aldrich, molecular weight 89,000–98,000) powder in 95 °C deionized water. The solution was stirred until transparent. Then the solution was spin-coated (3000 rpm, 60 s) on the sample, and another ITO glass as the top LC contact. The coated sample is annealed at 55 °C for two hours to remove water from the polymer.

Next, we use an electron-beam evaporator to deposit ~2 nm SiO₂ on top of the PVA layer. Specifically, the substrate was tilted to 80° with respect to the normal deposition plane[69]. The high-angle deposition creates nanoscale structures used to align the liquid crystal. Glass microbeads at a diameter of 4.3 $\mu m$ were dispersed in IPA (10 mg/L, ultrasonicated for an hour) and spin-coated (4000 rpm, 40 s) on the alignment layer as LC spacers. A drop of liquid crystal, 4-cyano-4′-pentylbiphenyl (5 CB) from Sigma Aldrich, was added to the chip. Then we overlapped the sample with the top ITO glass slide with a PVA alignment layer on it. The edges of the two glass slides were sealed with epoxy. Large metal pads stretching out of the sealing region were fabricated and used to apply the supply voltage $V_{dd}$.

### Transmission characterizations
The LC modulator without the TPT layer was characterized for selecting the operation conditions. We illuminated a 1 cm × 1 cm large uniform device with a thermal lamp and measured the transmitted illuminance with a lux meter. We observed an unstable threshold shift for DC voltage, which is related to ion and water contamination of the LC[70], so mobile ions can flow inside the LC and partially screen the modulation electrical field. To enable stable, repeatable LC modulation performance, we used a rapidly changing square wave at 500 Hz to modulate the LC (and also for the $V_{dd}$ in TPT-LC stack measurements, with $V_{dd}$ representing the peak-to-peak voltage). The AC field is too fast

to cause slow contamination ion drift in LC, but is still sufficient to align the liquid crystal[71]. The same AC measurement scheme was applied to the measurements in Figs. 3–5. The measurement result of the pure LC modulator is shown in Supplementary Fig. 3.

The transmission ratios were calculated as the ratio between the transmissions at the specific test condition ($I_{test}$) over the reference high transmission at $V_{dd} = 0$ V under weak light ($I_{ON}$), as reported in Fig. 3 captions:

$$T_R = \frac{I_{test}}{I_{ON}}$$

such that $I_{ON}$ was measured at $V_{dd} = 0$. The only exceptions for this are Fig. 3b, when the two polarizers are parallel and the transmission is low at 0 V; for that case, we defined the reference high transmission $I_{ON}$ as the transmission at a very high voltage under weak light, i.e., $V_{dd} = 16$ V was used. Our tested device's transmittance already converges to a maximized constant below this voltage for parallel polarizers, indicating it as a feasible definition of the reference high transmission for the parallel polarizer configuration.

The change in phase was also studied with a polarization and phase-sensitive spectroscopy method[72], with an extra 90° rotation in the transmission matrix due to the twisted LC. The extracted parameters are used to calculate the slight phase modulation accompanying the amplitude modulation, as shown in Supplementary Fig. 3b. The phase modulation is relatively weak, at -0.06 rad/V.

### Light source quantification

The thermal lamp and the white LED were measured with a spectroscopy system (Horiba Scientific). The light was fed into the system through the objective lens and filtered by the grating system to resolve the spectra. The thermal lamp was kept at a fixed power during these measurements and the nonlinear transmission tests. An attenuator was applied to control the intensity projected on the sample during the transmission test. This avoids possible spectral shifts when the thermal lamp filament is at different temperatures. The luminous efficiency of the lamp was calculated by integrating spectrally resolved luminous efficiency with the lamp spectrum. The photonic luminous efficiency is 68 lm/W. The result is comparable with the previously reported values: black body spectrum at 4000 K peaks at 700 nm and has a luminous efficiency of 55 lm/W[73]. Our lamp spectrum also peaks -700 nm but has a much smaller infrared tail, probably due to infrared absorption on the chamber walls of the torch, leading to a higher luminous efficiency than the 4000 K black body radiation. The white LED has a luminous efficiency of 200 lm/W.

### Photoresponse measurements

The photoresponse at 633 nm and 488 nm was measured with the Horiba Raman spectroscopy system. We used the CCD camera in the Horiba system to adjust the laser spot size and overlap it with the device pixel. The incident powers on the device were calibrated with a power meter placed under the objective lens. The device uniformity (Fig. 2f) was estimated over 116 devices that are fabricated on four chips with identical process flow and separately wired out. On each chip, 29 devices were tested, spanning an area of 14 mm × 10 mm surrounding the arrayed TPTs. The light resistances were measured under a thermal lamp at an illuminance of $1.4 \times 10^4$ lux. The thermal lamp spectrum is shown in Supplementary Fig. 6, with a center wavelength of 713 nm and a full-width-at-half-maximum (FWHM) bandwidth of 175 nm.

### Glare reduction analysis

We evaluated the image quality with and without glare reduction in the five boxes shown in Fig. 4b. The image in each box was transformed into an 80 × 50 matrix ($M_{jk}$) by summing up the RGB values.

We calculated the mean intensity of the $N = 50$ image pixels along the direction of the grating lines to evaluate the averaged pattern ($\bar{I}_j$):

$$\bar{I}_j = \frac{1}{N} \sum_{k=1}^{N} M_{jk} \tag{1}$$

This vector, $\bar{I}_j$, is plotted with solid lines in Fig. 4b–f. The corrected sample standard deviations ($s_j$) along the 50 image pixels were also evaluated for each column and plotted as the error bars in Fig. 4b–f:

$$s_j = \sqrt{\frac{1}{N-1} \sum_{k=1}^{N} \left( M_{jk} - \bar{I}_j \right)^2} \quad j = 1,2,\ldots,80; \quad N = 50 \tag{2}$$

We defined the image signal as the difference between the average intensity in the bright region ($R_{bright}$, with $N_{bright}$ pixels) and the dark region ($R_{dark}$, with $N_{dark}$ pixels) of the grating lines. The bright/dark regions were selected as the pixels with the top/bottom 25% intensities in the dynamic range, which was defined by the maximum intensity minus the minimum intensity in the box. The bright/dark regions were only segmented once using the no-glare image in Fig. 4b, and applied to all subsequent SNR evaluations at different glare, voltage, and exposure settings. The SNR was calculated as the ratio of the signal over the standard deviation in the dark regions:

$$\bar{I}_{bright} = \frac{1}{N_{bright}} \sum_{jk \in R_{bright}} M_{jk} \tag{3}$$

$$\bar{I}_{dark} = \frac{1}{N_{dark}} \sum_{jk \in R_{dark}} M_{jk} \tag{4}$$

$$SNR = \frac{\bar{I}_{bright} - \bar{I}_{dark}}{\sigma} \tag{5}$$

$\sigma$ is the corrected sample standard deviation of all the data points within the dark regions in the box:

$$\sigma = \sqrt{\frac{1}{N_{dark} - 1} \sum_{jk \in R_{dark}} (M_{jk} - \bar{I}_{dark})^2} \tag{6}$$

where $N_{dark}$ is the total number of image data points in the dark regions of the grating lines.

The pattern contrast $C$ (plotted in Supplementary Fig. 7) is evaluated as below:

$$C = \frac{\bar{I}_{bright} - \bar{I}_{dark}}{(\bar{I}_{bright} + \bar{I}_{dark})/2} \tag{7}$$

The temporal evolutions of intensity in Fig. 5 were evaluated with summed RGB values over the displayed region (darker lines in the curves) and region marked by the red box in Fig. 5a.

### Data availability

The data that support the plots within this paper and other findings of this study are available from the corresponding authors upon request.

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

## Acknowledgements

We acknowledge the California NanoSystems Institute (CNSI) at UCLA for the device fabrication and technical support. The project is supported by the UCLA CNSI Noble Family Innovation Fund. X.D. acknowledges partial support from the Office of Naval Research through grant no. N00014-22-1-2631 for 2D device fabrication and optimizations. A provisional patent application on the optoelectronic neuron array has been filed with the United States Patent and trademark office (see Competing Interests).

## Author contributions

X.D., A.O. and D.Z. conceived the research. D.X., J.Z., and Y.Z. prepared the VDWTF material. D.Z., D.X., J. H., P.W., X.L. and L.W. performed device fabrication. D.Z., Y.Li, Y.Luo, B.Z., H.R., B.B. and A.Z. performed the device characterization and data analysis. D.Z., M.J., Y.H., A.O. and X.D. co-wrote the manuscript. All authors discussed the results and commented on the manuscript.

## Competing interests

The authors X.D, A.O, and D.Z., affiliated with the University of California, Los Angeles (Los Angeles, California), are inventors of a United States Provisional Patent Application (# 63/497,969) on the optoelectronic neuron array. The remaining authors declare no competing interests.
