## [Peer Review File · Nature Communications]

Broadband nonlinear modulation of incoherent light using a transparent optoelectronic neuron arrayEditorial Note: This manuscript has been previously reviewed at another journal that is not operating a transparent peer review scheme. This document only contains reviewer comments and rebuttal letters for versions considered at *Nature Communications*. Mentions of the other journal have been redacted.

Reviewer #1 (Remarks to the Author):

Although I don't think it is deserved for [redacted], it aligns well with the scope of Nature Communications. Therefore, I would like to recommend its publication in its current form on Nature Communications.

Reviewer #4 (Remarks to the Author):

I have carefully reviewed the revised manuscript titled "Broadband nonlinear modulation of incoherent light using a transparent optoelectronic neuron array" and the associated responses to the previous reviewers' comments. I appreciate the thorough efforts the authors have made to address the concerns raised during the previous review process.

The revisions made by the authors have significantly strengthened the manuscript. The clarity of the presentation, the scientific rigor, and the overall structure have notably improved. The experimental design and results are now more robust and effectively address the previously identified issues.

The authors have demonstrated a commendable understanding of the feedback received, and their revisions have successfully enhanced the manuscript's quality. The study's originality and its contributions to the field of optoelectronics are now more evident.

I recommend accepting the manuscript for publication in Nature Communications. The research is well-conducted, and the findings are of significance to the scientific community. The manuscript aligns with the high standards of the journal, and its publication would be a valuable addition.

Reviewer #5 (Remarks to the Author):

Zhang et al. have successfully developed an optoelectronic nonlinear filter array that can precisely adjust contrast across different wavelengths. Impressively, this array, with a large number of 10,000 pixels, can quickly reduce glare while keeping low-light objects visible in a camera's field of view. This work is quite impressive, and the results could be widely used in various industries, showing the great potential of the array. However, despite improvements from several revisions, there are still some important issues that need to be addressed for the paper to be suitable for publication in Nature Communications.

1) The term "optoelectronic neuron array" used in the paper doesn't seem to be correct. Normally, a "neuron" should combine several inputs and produce an output based on a specific rule (transfer function). Therefore, in order to call the optoelectronic "filter" a "neuron array", it must be shown that it can combine each light signal and produce an "output" according to a certain rule (transfer function), and describe how this could work in an optical neural network. If this isn't explained, the term should be changed to "optoelectronic filter array".

2) The main focus of the theory in the paper is to explain how each pixel works. To make the research more broadly useful, it's important to thoroughly explain the role of the liquid crystal (LC) layer, which is key to creating the nonlinear optical response seen. In Figure 1d, the LC layer appears to change how light passes through by arranging crystals of different sizes when an electric field (E-field) is applied. The arrangement and the change in light transmission depend on the strength of the electric field. This leads to questions about the size of the crystals in the LC layer, the strength of the electric

field needed to arrange the crystals, and the basic principles behind this arrangement with optical simulation(such as FDTD simulation).

Response to Reviewer #1:

Although I don't think it is deserved for [redacted], it aligns well with the scope of Nature Communications. Therefore, I would like to recommend its publication in its current form on Nature Communications.

Response: We thank the reviewer for taking the time reviewing our paper and providing helpful feedback.

Response to Reviewer # 4:

I have carefully reviewed the revised manuscript titled "Broadband nonlinear modulation of incoherent light using a transparent optoelectronic neuron array" and the associated responses to the previous reviewers' comments. I appreciate the thorough efforts the authors have made to address the concerns raised during the previous review process.

The revisions made by the authors have significantly strengthened the manuscript. The clarity of the presentation, the scientific rigor, and the overall structure have notably improved. The experimental design and results are now more robust and effectively address the previously identified issues.

The authors have demonstrated a commendable understanding of the feedback received, and their revisions have successfully enhanced the manuscript's quality. The study's originality and its contributions to the field of optoelectronics are now more evident.

I recommend accepting the manuscript for publication in Nature Communications. The research is well-conducted, and the findings are of significance to the scientific community. The manuscript aligns with the high standards of the journal, and its publication would be a valuable addition.

Response: We thank the reviewer for taking the time to review our paper and for the very supportive comments.

Response to Reviewer #5:

Zhang et al. have successfully developed an optoelectronic nonlinear filter array that can precisely adjust contrast across different wavelengths. Impressively, this array, with a large number of 10,000 pixels, can quickly reduce glare while keeping low-light objects visible in a camera's field of view. This work is quite impressive, and the results could be widely used in various industries, showing the great potential of the array. However, despite improvements from several revisions, there are still some important issues that need to be addressed for the paper to be suitable for publication in Nature Communications.

Response: We thank the reviewer for taking the time to review our paper, and for the supportive comments.

[Point 1] *The term "optoelectronic neuron array" used in the paper doesn't seem to be correct. Normally, a "neuron" should combine several inputs and produce an output based on a specific rule (transfer function). Therefore, in order to call the optoelectronic "filter" a "neuron array", it must be shown that it can combine each light signal and produce an "output" according to a certain rule (transfer function), and describe how this could work in an optical neural network. If this isn't explained, the term should be changed to "optoelectronic filter array".*

Response: Thanks for the very helpful comment. To clarify this point, we have added the following text to our revised Introduction section:

When coupled with free-space optical computing systems, composed of, for example, task-specific diffractive processors, the optical inputs to this nonlinear optoelectronic neuron array can take programmable convolutional or fully-connected filters, covering any desired set of spatially varying point spread functions between the visual scene and the plane of the optoelectronic device. Therefore, cascaded integration of optoelectronic neuron arrays with linear diffractive processors could be used to construct nonlinear optical networks, potentially

finding widespread applications in computational imaging and sensing, also opening the door for new nonlinear optical processor designs using ambient light.

[Point 2] *The main focus of the theory in the paper is to explain how each pixel works. To make the research more broadly useful, it's important to thoroughly explain the role of the liquid crystal (LC) layer, which is key to creating the nonlinear optical response seen. In Figure 1d, the LC layer appears to change how light passes through by arranging crystals of different sizes when an electric field (E-field) is applied. The arrangement and the change in light transmission depend on the strength of the electric field. This leads to questions about the size of the crystals in the LC layer, the strength of the electric field needed to arrange the crystals, and the basic principles behind this arrangement with optical simulation (such as FDTD simulation).*

Response: Thanks for the comment. All the liquid molecules are anisotropic, rendered as ellipsoids. They have the same size in the LC layer but take different molecular orientations. The molecules gradually rotate their orientations in the light propagation direction to form a spiral pattern, rotating the incident light's polarization. The concept is borrowed from the twisted nematic LCD technology, which is well-studied both optically and electrically in previous works such as Ref [63] and [68].

Indeed we are **not** changing the crystal shape - instead aligning their molecular orientation. To better clarify this point, we have added the following sentence to the captions of Fig. 1:

“The liquid molecules are anisotropic, rendered as ellipsoids. The molecules gradually rotate their 3D orientations to form a spiral pattern, rotating the incident light's polarization.”